# Performance Pay in Hospitals: An Experiment on Bonus–Malus Incentives

**DOI:** 10.3390/ijerph17228320

**Published:** 2020-11-10

**Authors:** Nadja Kairies-Schwarz, Claudia Souček

**Affiliations:** Faculty of Economics and Business Administration & CINCH—Health Economics Research Center, University of Duisburg-Essen, 45127 Essen, Germany; claudia.soucek@wiwinf.uni-due.de

**Keywords:** pay for performance, diagnosis related groups, bonus–malus incentives, artefactual field experiment, laboratory experiment

## Abstract

Recent policy reforms in Germany require the introduction of a performance pay component with bonus–malus incentives in the inpatient care sector. We conduct a controlled online experiment with real hospital physicians from public hospitals and medical students in Germany, in which we investigate the effects of introducing a performance pay component with bonus–malus incentives to a simplified version of the German Diagnosis Related Groups (DRG) system using a sequential design with stylized routine cases. In both parts, participants choose between the patient optimal and profit maximizing treatment option for the same eight stylized routine cases. We find that the introduction of bonus–malus incentives only statistically significantly increases hospital physicians’ proportion of patient optimal choices for cases with high monetary baseline DRG incentives to choose the profit maximizing option. Medical students behave qualitatively similar. However, they are statistically significantly less patient oriented than real hospital physicians, and statistically significantly increase their patient optimal decisions with the introduction of bonus–malus incentives in all stylized routine cases. Overall, our results indicate that whether the introduction of a performance pay component with bonus–malus incentives to the (German) DRG system has a positive effect on the quality of care or not particularly depends on the monetary incentives implemented in the DRG system as well as the type of participants and their initial level of patient orientation.

## 1. Introduction

Over the last decade performance pay incentives have gained increasing relevance to raise the quality of health care in the inpatient care sector [1,2]. Nowadays, the majority of OECD countries employs some type of performance pay component mainly with a focus on rewarding good healthcare quality, see e.g., the Premier Hospital Quality Incentive Demonstration Project in the USA [3]. However, the effects of performance programs with bonus payments on the quality of care are, if at all present, rather modest and only temporary [4,5,6,7,8,9,10,11,12]. 

More recent performance programs also make use of malus incentives. Malus payments, or penalties, are negative payments for poor healthcare quality. The latter are often implemented in the form of fines that have to be paid back or only partial payments, e.g., reduced reimbursement fees, in case of poor performance. The idea of malus incentives relies on the behavioral concept of (cumulative) prospect theory [13,14], which assumes that losses loom larger than gains. This implies malus incentives inducing individuals’ loss aversion might intensify the effectiveness of performance incentives more than malus payments. However, the effects of programs with pure malus incentives, which were implemented by some countries such as Denmark, England, and the USA on the quality of care are also, if at all present, rather modest and only temporary [15,16,17,18,19,20].

A possible alternative are performance pay components combining bonus and malus incentives. Germany has been legally committed to introducing such a performance pay component in the inpatient care sector [§5 Hospital Remuneration Law/Krankenhausentgeltgesetz—KHEntgG]. However, the evidence on the effect of a combined performance pay component with bonus–malus incentives on the quality of care is scarce. There is only limited evidence on the effects of a combination of both types of performance pay components, as aspired by the German government, from South Korea. Here, the hospital remuneration with fee-for-service was directly replaced by a Diagnosis Related Groups (DRG) system with bonus–malus incentives based on treatment quality. The latter are modeled in a way that bonuses are paid to hospitals with superior healthcare quality while reimbursement fees are lowered for poor performing hospitals [21,22,23]. The few studies to evaluate this program show positive effects of the introduction of such bonus–malus incentives, i.e., increased quality of care and reduced medical costs. However, given that the fee-for-service was replaced by a DRG system with bonus–malus incentives based on treatment quality, disentangling the effect of introducing a DRG system from the one of bonus–malus incentives is difficult.

The objective of this paper is to provide first controlled evidence on the effects of introducing a performance component with bonus–malus incentives to a DRG system on physician provision behavior in the inpatient care sector. For this, we conduct an online experiment with medical students of higher semesters and hospital physicians. Similar to the framed field experiment by Eilermann et al. [24], participants in the experiment are confronted with stylized routine cases that have previously been validated by real physicians. Choices are binary in the sense that there is one profit maximizing and one patient optimal treatment alternative. The experiment uses a sequential design resembling the introduction of a performance pay component with bonus–malus incentives to a simplified version of the German DRG system. While the payment system modelled in the experiment resembles the German one, it is also similar to systems in many other countries, see, e.g., Australia, England, France, or Netherlands.

Our results show that given a simplified version of the current German DRG remuneration system in part 1, hospital physicians choose the patient optimal alternative, which leads to the highest benefit for the patient, in 74% of all treatment choices. While the introduction of a performance pay component with bonus–malus incentives does not lead to an overall statistically significant increase in the proportion of patient optimal choices, we find statistically significant changes towards more patient optimal choices for cases with high monetary baseline DRG incentives to choose the profit maximizing alternative. Particularly for the latter cases, the introduction of a performance pay component yields a decrease in the monetary amount physicians have to give up to choose the patient optimal alternative. Even though medical students behave qualitatively similar to hospital physicians, they choose statistically significantly less patient optimal alternatives under the simplified DRG system in part 1, and statistically significantly increase the proportion of patient optimal choices for all stylized routine cases with the introduction of the bonus–malus incentives in part 2. These results indicate that whether the introduction of a performance pay component with bonus–malus incentives to the (German) DRG system has a positive effect on the quality of care or not particularly depends on the monetary incentives implemented in the DRG system as well as the type of participants and their initial level of patient orientation. Our experimental design may serve as a wind tunnel study and proposes that more research is needed to determine effective performance incentives to achieve better quality across all therapeutic areas and patient cases.

The paper proceeds as follows. A literature review and main research questions are provided in Section 2. The design of the experiment is given in Section 3. The results are presented in Section 4 and lead to the discussion and conclusion in Section 5.

## 2. Literature Review and Research Questions

First, we aim at investigating physicians’ baseline treatment behavior given a simplified version of the current German reimbursement system using DRGs in the hospital sector. In the current system all hospital patients are assigned to a DRG based on a grouping algorithm, which includes the coded diagnosis among other things, such as gender, age, or comorbidity risk. Each DRG has its own cost weight, which is multiplied with a base rate resulting in the respective DRG fee per case, which is very similar to a capitation remuneration. The DRG fees hospitals receive as reimbursement are based on the average costs of a sample of hospitals representative for the hospital landscape in Germany. Hence, the characteristics of a hospital with regards to their specialization and cost structure affect the profit a hospital can earn from the DRG fees. This principle leads to incentives of reducing costs per patient in order to increase the revenue per patient given that not all services are paid as is the case in a fee-for-service system. 

Theoretical research indicates that this specific design of a DRG system leads to incentives of over- or under-provision of patients [25]. Specifically, this means that in DRG systems in which the price is based on average costs the refinement of a DRG, i.e., splitting one DRG into several based on the treatment the patient receives, leads to overprovision of the more intensive treatment option, whereas having one DRG for several more or less intensive treatments of the same diagnosis leads to under-provision of the more intensive treatment option. Hence, whether refinement of a DRG is welfare optimizing depends on the benefit and cost function of each hospital. Further theoretical evidence also finds that when accounting for the characteristics of the German healthcare system such as the principle of “benefits in kind”, i.e., insured people in statutory health insurances do not have to pay the medical costs in advance as opposed to the principle of cost reimbursement in private health insurances, such a remuneration system does not yield optimal results in terms of quality [26,27]. Evidence from the field supports these theoretical findings [28,29,30,31,32]. Furthermore, the introduction of DRG systems is prone to a number of different behavioral aspects such as upcoding patients to for instance a higher risk group in order to receive a higher capitation payment [33,34,35]. Thus, it is difficult to control for all aspects influencing physician provision behavior. To gain control, we abstract from the possibility to upcode in this paper. 

Research question 1: *How do hospital physicians provide medical treatment in a simplified German DRG system?*

Second, we intend to investigate the effect of introducing a performance component with bonus–malus incentives to the simplified DRG system on hospital physicians’ treatment choices. There is vast empirical evidence on existing performance components such as Rosenthal and Frank [4] who summarize the empirical evidence on performance pay in the health care sector, including both in- and outpatient care, as well as comparable interventions in other sectors. They find scarce evidence for the effectiveness independently of the sector. Further research focusing solely on bonus incentives in the health care sector, comes to a similar conclusion while stressing the difficulty of disentangling the effect of the performance component from other jointly introduced measures [15,36,37]. Performance pay components are for instance frequently introduced jointly with other policy interventions such as public reporting [21,36,37,38,39]. Further, the design and incompleteness of performance measures often results in gaming of performance indicators or multitasking [40,41,42]. The difficulties to identify causal effects suggest controlled laboratory experiments to be well suited as complementary research. The existing studies using controlled laboratory experiments with medical students find positive effects of the introduction of a performance pay component with a bonus on physician treatment behavior [43,44,45,46]. 

Moreover, there is only little evidence on the effects of performance components with malus incentives and even fewer evidence on their effects on the quality of care [15,16,17,18,19,20]. While there are no laboratory experiments on the effects of malus incentives on physician provision behavior yet, experiments in other areas on malus incentives come to the conclusion that penalties increase the performance of the participants [47,48,49,50]. Furthermore, the outline of the planned German performance component with bonus–malus incentives is unique in the sense that the hospitals are not supposed to receive a bonus or pay a penalty, but to receive a higher or reduced DRG fee. The only country that implemented a performance component with bonus–malus incentives is South Korea. While the latter has not been analyzed as extensively as other performance components using bonus or malus incentives only, the evidence finds positive effects on the quality of care as well as on the efforts to reduce medical costs [21,22,23]. However, there is no controlled analysis of the effects of introducing bonus–malus incentives to a DRG system like the German one. 

Research question 2: *Does the provision behavior of hospital physicians change with the introduction of performance component with bonus–malus incentives to a simplified German DRG system?*

Third, we aim to analyze whether there are any behavioral differences between real practicing hospital physicians and medical students and if so, of which type. Harrison and List [51] argue that a more realistic subject pool, i.e., in this case real physicians, might behave differently than students in the experiment since the former not only decide based on the information provided in the experiment but also based on their real-world experience. Several experimental studies hence also use real physicians as participants to investigate how they respond to payment incentives [24,45,52,53]. However, to the best of our knowledge, only two of these studies include real physicians as well as medical students to investigate the effects of payment incentives on physician provision behavior in controlled experiments [45,52]. Their results show that both physicians and medical students are influenced by the monetary incentives of the remuneration systems. Both find the same qualitative results for medical students and real physicians. While Brosig-Koch et al. [45] show that real physicians react less to the respective payment incentive and are statistically significantly closer to the patient optimal treatment levels, Reif et al. [52] find almost no statistically significant differences. However, the experimental designs only allow for across subject comparisons. Within subject comparisons for a change in remuneration such as the introduction of a performance component are not studied. Given the result that physicians react less to the payment incentives and are per se closer to the patient optimal levels, changes due to performance pay might be smaller for physicians than for medical students. 

Research question 3: *Is there a difference in treatment behavior between practicing hospital physicians and medical students? How does this affect the introduction of a performance component with bonus–malus incentives to a simplified German DRG system?*

## 3. Experimental Design

The objective of the experiment is to analyze whether the introduction of a performance pay component with bonus–malus incentives leads to more patient optimal treatment decisions, and hence improves the quality of healthcare. To investigate the effects of introducing such a performance pay component, we implemented a sequential design with a medical framing and stylized routine cases. The experiment consisted of two parts. While participants were confronted with a simplified version of the current remuneration for German hospitals with DRG fees in the first part, they were introduced to an additional performance pay component with bonus–malus incentives in the second part. Moreover, note that in order to investigate the effects of the introduction of a performance pay component with bonus–malus incentives, the decision environment abstracted from other certain real world aspects like uncertainty, e.g., in the form of legal risks connected to the treatment path, which could potentially affect treatment behavior. 

### 3.1. Treatment Cases

In the computer-based experiment, each participant decided as a hospital physician on how to treat the same eight patients resembled by stylized routine cases, see, e.g., Sherry [54] for a similar approach for pediatricians. Hence, the experiment did not involve real patients. 

For the stylized routine cases we explicitly chose one medical field that had been discussed as area of concern regarding health care quality and one that has not. For the former we chose cardiology since the disproportionate increase in inpatient cases for expensive interventional procedures and its negative effect on health care quality of the patients have been vividly discussed [55] (p. 135). For the latter we decided on diabetology, as it overlaps content related with cardiology but differs regarding quality concerns. The stylized routine cases we implemented were set up with the help of cardiologists and diabetologists in both medical fields. Particularly, the eight specific cases in both fields were chosen by clinical experts in cardiology and diabetology based upon the fact that they pose a clear trade off decision between a patient optimal and profit maximizing option in the inpatient setting in Germany. This does not mean that the profit maximizing option harms the patients; however, the patient optimal alternative is the one which was recommended based on systematic research of all available evidence summarized in evidence-based guidelines. Hence, a deviation from the evidence-based guidelines may also reflect a different expert opinion.

For each case, physicians received simplified medical information about the condition. In addition to this information they were presented two treatment options (see Figure 1 for an example). The structure of the cases as well as the treatment options were based on case studies used in medical courses at university as well as further trainings for physicians enabling the participants to easily grasp the provided information without being too simplified. They were also reviewed by cardiologists and diabetologists with regard to medical correctness and suitability for an experiment reaching a broader range of participants. The stylized routine cases varied by degree of severity, half of them being moderate and the other half being severe cases. For each case, one option was clearly identifiable as patient optimal and the other as profit maximizing (see Figure 1). Lastly, the stylized routine cases differed in the level of monetary incentives to choose the profit maximizing alternative, where half of the cases are low incentive and the other half high incentive cases. The level of monetary incentive was determined based on the real world DRG fee differences between the patient optimal and the profit maximizing option for each case. The numbers can be found in Table A1, Table A2, Table A3, Table A4, Table A5 and Table A6.

To ensure that medical students also understood the cases, the subject pool was restricted to higher semesters (at least one year of medical studies). To this end the majority of them should have had dealt with both medical fields in the course of their studies. We also asked for their experience with both of these fields in the ex post questionnaire. Irrespective of this restriction, participants were shown the evidence-based guideline recommendation for each case, and thus could clearly identify the patient optimal alternative. According to ex ante feedback from physicians specialized in diabetology or cardiology, the guidelines in the experiment present valuable sources of treatment recommendations. Due to legal constraints, patients in our baseline treatment with real hospital physicians are only abstract and patient benefits are not presented in monetary amounts, and thus not transferred to real patients outside the lab as it has become a standard in related medically framed experiments [45,56,57,58,59,60]. Therefore, it is possible that physicians choose only the profit maximizing alternative since they do not harm real patients but do receive a real payment based on their decisions. The incentive for choosing the patient optimal alternative is altruism or conformity to guidelines given the patient optimal alternative is the option recommended by evidence-based guidelines for this specific patient type. Our results for physicians’ patient orientation should thus be rather conservative. However, we control for this with two experimental conditions with medical students in which one resembles the baseline condition and the other includes monetary incentives for choosing the patient optimal alternative. In the latter students are also presented patient benefits in monetary amounts that go to the charity Christoffel Blindenmission to treat real patients with eye cataract. 

### 3.2. Payment Incentives

We follow the guidelines of economic experiments in which the decision environment is designed such that the payoffs motivate participants’ motivations for their choices. Participants in the experiment are paid out at the end of the experiment. How much they earn depends on their decisions made in the experiment. As previously described, the decisions for the eight stylized routine cases in our setup involved a monetary trade-off between own profit and (monetary) patient benefit. Particularly, when choosing the profit maximizing option the physician earns the maximum payoff while the (monetary) benefit for the patient is less than for the patient optimal choice. In contrast, when choosing the patient optimal option, the physician earns less than the maximum, while the patient benefit is the highest possible one. By choosing medical treatment options, i.e., either profit- or patient-benefit maximizing, participants thus reveal their motivation and determine their own payoffs. 

In the first part of the experiment, the remuneration resembled a simplified German hospital remuneration with DRG fees. The remuneration information within the experiment consists of several components given that the physicians in real-world do not earn the reimbursement directly. Therefore, the stylized routine cases include the reimbursement the hospital earns in form of the DRG fee, the costs, and the resulting profit for the hospital as well as the remuneration for the participants. The latter was a fixed fee that varied by the degree of severity and the treatment alternative (either patient optimal or profit maximizing), and hence is in line with the DRG fee mechanism and real-world incentives for chief physicians given that their contracts include increasingly more performance based components regarding their department´s budget. This is certainly not the case for other hospital physicians yet. However, treatment guidelines at hospitals are mostly determined by chief physicians, and therefore these incentives indirectly exist for other physicians as well. This reason was stated by most cardiologists and diabetologists with whom we worked on the treatment cases and the design of the experiment. The DRG fees implemented within the experiment as well as the differences between them for different treatment alternatives as well as moderate and severe cases were determined by the valid rates in Germany from 2015 [61]. For an overview of the monetary parameters see Table A1, Table A2, Table A3, Table A4, Table A5 and Table A6. 

The amounts for participants´ average expected payoffs were moreover set based on the average hourly wage for physicians and students, i.e., physicians could earn a fixed fee between €3 and €15 and students between €0.33 and €3. For physicians we assumed an average hourly wage of €150 and for students €12. To capture the specificities of a real chief hospital physician’s payment, participants´ remuneration also comprised a compensation directly linked to the hospital budget that was determined by the budget impact of all the chosen options by each participant [62]. Here, the budget impact was the difference between the DRG fees and costs for the chosen treatment option. The latter had been estimated by medical controllers working in German hospitals. Depending on whether the total budget impact was negative or neutral/positive, participants could earn a lump sum remuneration. The amount of this lump sum varied the same way as the fixed remuneration of the first part of the remuneration, so again between €3 and €15 for physicians and €0.33 and €3 for students. 

In the second part of the experiment, the performance pay component with bonus–malus incentives was introduced. We kept the monetary incentives regarding the budget impact of the chosen options constant across both parts of the experiment. Hence, part of the remuneration was also the lump sum. However, the new part of the remuneration did not depend on each chosen option anymore but was determined by a quality indicator as is stipulated in the structure of the planned German performance pay component (see Figure 2 for a summary of the design). 

Hence, we introduced quality indicators for each respective treatment option. Given that the final structure of the quality measurement for the German performance pay component had not been finally defined by the point of design, the quality scores in the experiment were based on the numeric quality measures of the US-American Premier Hospital Quality Incentive Demonstration Project. The latter had been used as a model for other systems such as the Advancing Quality Program [3]. For simplification we defined fixed values for each quality indicator of a treatment option. Thus, we abstracted from uncertainty regarding the impact of a chosen treatment option on the overall quality. The values of the quality indicators did neither depend on the treatment case nor on the degree of severity as this would have induced a discrimination towards the more severe cases. Hence, the values varied according to the patient optimal and profit maximizing option, i.e., if the participant chose the profit maximizing option, the values were considerably lower (between 10% and 26%).

At the end of the experiment, the quality indicators for all of the eight chosen options were summarized in a total quality score in the form of an arithmetic average preventing the occurrence of the multitasking problem. The total quality score then determined the new part of the remuneration for part 2 that varied the same way as the other compensation components. The outline of the German performance pay component stipulates that the hospitals receive a percentage bonus or malus on their regular DRG fee, i.e., they either receive a higher or a reduced DRG fee depending on the quality score. The penalty is to be twice as high as the bonus [§5 KHEntgG]. However, it is not further stated whether the bonus or malus is applied based on an absolute quality score threshold or if they depend on the relative performance to other hospitals. For simplification, the performance pay component was hence conditioned upon exceeding or falling below an absolute threshold.

The range of the threshold values were determined based on the examples of the US-American performance pay component [63]. A total quality score of 100 represented an average quality (i.e., €11 for physicians and €2.50 for students), larger than 104 above average quality (i.e., €15 for physicians and €3 for students), and smaller than 96 below average quality (i.e., €3 for physicians and €1.50 for students). Depending on the individual total quality score, participants could earn up to the same maximum amount of remuneration as in the first part of the experiment with the fixed DRG fee. However, in order to do that they had to choose more patient optimal options. For a more formal description of the payment incentives see Section A.1. 

### 3.3. Experimental Protocol

The experiment was programmed with oTree [64] and conducted online for both subject pools. The authors conducted their experimental study with medical students via the Essen Laboratory for Experimental Economics and committed themselves to the rules of this laboratory before conducting their experiment. The rules of the Essen Laboratory for Experimental Economics conform to the Ethics Directive of the German Association for Experimental Economic Research e.V. Note, that this was regarded as sufficient by the head of the laboratory as no real patients were involved. To be eligible for participating in the experiment, physicians either had to be in training to gain the cardiology or diabetology specialty or were already experienced specialists in one of the medical fields; either way all had to practice in their specialty. While we recruited a total of 35 physicians working in public hospitals via email or telephone, only 16 completed the experiment. For more details on the recruiting process for hospital physicians and medical students see Section A.2. They conducted the experiment without the donations to real patients outside the lab. As the number of hospital physicians is rather small, we also recruited 40 medical students of second or higher semesters using the online recruiting system ORSEE [65]. Of the latter 19 conducted the identical experiment as the physicians and the other 21 the experiment with donations. For an overview of treatment conditions see Table 1, and for a more detailed description of sample characteristics see Table 2.

The experimental procedure was identical for both subject pools and treatments. They were sent a link to the experiment which opened in a web browser. All steps were predefined by the program and the participants could decide on their own how long the experiment lasted. The experiment began with the instructions in which a telephone number could be contacted for help with clarifying questions (see Section A.3 for Instructions); however, no one asked questions. In order to check the comprehensibility of the design, especially of the compensation elements, the experiment was tested beforehand and participants within the experiment also had to answer a compensation question before making the treatment decisions. Then, each participant had to choose between two treatment options for eight stylized routine cases that were used for both parts of the experiment. All of the cases were displayed on one page and the order was predefined and the same for all. After the experiment participants were paid via a bank transfer. To verify the donation, the medical students within the respective treatment received a receipt of the bank transfer for the total donation via email.

Sessions lasted on average 34 min for the physicians and 40 and 51 min for the medical students (51 min for the treatment with donations). The physicians earned an average amount of €58.00 while the medical students received €10.87 for the identical experiment. The medical students in the treatment with donations received with €8.85 slightly less. In total, €40.49 were transferred to the Christoffel Blindenmission in the latter treatment condition. 

## 4. Results

### 4.1. Physician Provision Behavior in the DRG System

First, we investigate treatment behavior of real hospital physicians given a simplified German DRG system in part 1 of the experiment. On aggregate, we find that hospital physicians choose the patient optimal option in 74% of all decisions. The proportion of patient optimal choices does not vary substantially and has a standard deviation of 16%. From Table 3 we can infer that in all stylized routine cases at least 50% of all hospital physicians follow the guideline recommendation and choose the patient optimal option. The lowest proportion of patient optimal choices can be observed for a moderate cardiological case (that is 50%) and the highest for the moderate diabetological case (that is 100%). The comparatively low fraction of patient optimal choices for the cardiological treatment case might be explained by treatment styles that differ from the medical guideline. In particular, this is the only case, which demands a decision between a drug therapy and an interventional procedure. The high proportion of physicians choosing the more expensive intervention might indicate a general preference for the use of interventional procedures for this type of case. This is also reflected by actual numbers showing that the latter are above average in Germany [66] (p. 82). Furthermore, this is one of the cases in which the treatment options are covered by different DRGs leading to an incentive to overprovide the intervention. This finding is in line the theoretical findings of Hafsteinsdottir and Siciliani [25]. Hence, the low proportion of patient optimal decisions might not only be explained by different treatment styles, but also by monetary incentives. 

Next, we investigate whether the medical field, degree of severity and level of monetary DRG incentive affect hospital physicians’ provision behavior. For this, we compare the distribution of the patient optimal and profit maximizing choices between stylized routine cases sorted by medical field, degree of severity and level of monetary DRG incentive across all physicians. For the medical field, we compare the distribution of patient optimal and profit maximizing decisions between diabetological and cardiological cases and do not observe statistically significant differences (*p* = 0.6865, Fisher´s exact test). When comparing moderate and severe cases, we do not observe statistically significant differences either (*p* = 1, Fisher´s exact test). The same applies to the comparison between levels of monetary DRG incentives (*p* = 0.4192, Fisher´s exact test). Finally, we analyze individual provision behavior. We find that 12% of the physicians are purely patient optimizing while the rest combines patient optimal and profit maximizing choices. 

**Result** **1.**
*Given a simplified German DRG remuneration, hospital physicians provide the patient optimal treatment in 74% of all cases. Neither the medical field, the severity of illness, nor the level of financial DRG incentives systematically affect their provision behavior.*


### 4.2. Impact of Bonus–Malus Incentives on Provision Behavior

Second, we aim to assess whether the introduction of a performance pay component with bonus–malus incentives to the DRG system changes the provision behavior of hospital physicians. On aggregate we find that this introduction leads to an increase in patient optimal choices from 74% to 84%. However, when comparing the distribution of patient optimal and profit maximizing decisions of part 1 with part 2, i.e., the DRG system with the DRG system including the bonus–malus incentives, across all stylized routine cases, we find no statistically significant differences (*p* = 0.4667, Fisher´s exact test). Moreover, we observe that 80% of total decision changes made are changes towards the patient optimal alternative, we also observe 20% of changes from the patient optimal to the profit maximizing alternative. This indicates that paying for performance may lead to motivation crowding out and confirms the results of former experiments [43,45].

Following the analysis of part 1, we continue by investigating changes in provision behavior by treatment case. Figure 3 illustrates the proportions of patient optimal behavior between part 1 and part 2 of the experiment. The highest proportion is now for a severe cardiological (100%) and the lowest for a severe diabetological case (69%). For the severe cardiological case two more patient optimal choices were offset by two decision changes to the profit maximizing option. The reason for that is the optimizing behavior of two physicians who chose the profit maximizing option in this case to compensate for budget losses due to decision changes to the patient optimal alternative in other cases. We also find that the variation decreases from a standard deviation of 16% to 11%.

Similar to part 1, we find no statistically significant impact of the degree of severity (*p* = 1, Fisher´s exact test) and medical field (*p* = 0.6355, Fisher´s exact test). However, when comparing the distribution of patient optimal and profit maximizing decisions in stylized routine cases with high monetary baseline DRG incentives between part 1 and 2, we do find statistically significantly more patient optimal choices with the performance pay component (*p* = 0.0146, Fisher´s exact test) in contrast to the low monetary baseline DRG incentive cases (*p* = 1, Fisher´s exact test). Hence, the performance pay component seems to work especially well in increasing the number of patient optimal treatments for cases in which the monetary baseline DRG incentives for not choosing the patient optimal alternative are high. Figure 3 also illustrates that the highest increase of 62% in the proportion of patient optimal decisions is in the second treatment case which is a moderate cardiological case with high monetary baseline DRG incentives to choose the profit maximizing alternative. As this is the case with the lowest proportion of patient optimal choices in part 1, it confirms that bonus–malus incentives work especially well for these cases. 

Finally, we investigate individual provision behavior. In contrast to part 1, the number of purely patient optimizing individuals quadruples from two (12%) to eight (50%). There continues to be no individual that is purely profit maximizing.

**Result** **2.**
*On aggregate, the introduction of a performance pay component with a bonus–malus incentive to a simplified German DRG system does not lead to a statistically significant increase in patient optimal behavior of hospital physicians. However, at treatment case level, we find that the bonus–malus incentives yield statistically significantly more patient optimal choices in cases with high monetary baseline DRG incentives to choose the profit maximizing alternative.*


### 4.3. Differences between Hospital Physicians and Medical Students

Third, we aim at substantiating our results. As hospital physicians specialized in cardiology or diabetology were extremely difficult to recruit, our sample size of 16 is rather small. Moreover, due to legal constraints we could only run the experiment without resulting benefits for real patients. To address these two issues, we conducted two additional treatments with medical students, i.e., one in line with the real hospital physicians’ condition without monetary patient benefits and one with benefits.

Comparing provision behavior of hospital physicians and medical students who conducted the identical experiment at the aggregate level, we find that while hospital physicians choose the patient optimal option in 74% of all decisions in part 1 under the simplified DRG system, medical students choose the latter in only 38% (see Table 3). The variation between cases is also 17% higher for the medical students with a standard deviation of 18%. Hence, when comparing the distribution of patient optimal and profit maximizing decisions between hospital physicians and medical students, hospital physicians choose statistically significantly more patient optimal alternatives than medical students (*p* < 0.0001, Fisher´s exact test). Nonetheless, except for the second treatment case (moderate cardiological case with high monetary baseline DRG incentives to choose profit maximizing alternative), we observe qualitatively similar behavioral patterns to hospital physicians. This might be explained by the fact that medical students might be less prone to have already formed individual treatment styles differing from the medical guideline. 

Furthermore, we are interested in whether medical students respond differently to the introduction of a performance pay component (see Table 4). We find that the proportion of patient optimal choices increases statistically significantly from 38% to 63% for medical students (*p* < 0.000, Fisher´s exact test). In two cases the proportion of patient optimal choices is even higher than for physicians (second case, i.e., moderate cardiological with high monetary DRG incentives and seventh case, i.e., severe diabetological case with low monetary DRG). Especially for the second case this is due to the bonus–malus incentive which counteracts the DRG monetary incentive. The variation between stylized routine cases also decreases by 15%. 

When investigating whether the medical field, degree of severity, and level of monetary incentive affect medical students’ treatment behavior within part 1 and 2, we find no statistically significant effects (for *p*-values see Table A9). However, similar to the subject pool of hospital physicians this changes when comparing the distribution of patient optimal and profit maximizing decisions in the high monetary baseline DRG incentive cases between part 1 and 2. Medical students choose statistically significantly more patient optimal alternatives in the high monetary baseline DRG incentive cases in part 2 (*p* = 0.0006, Fisher´s exact test). Furthermore, they also react statistically significantly, albeit less, intensively to the bonus–malus incentives in the low monetary baseline DRG incentive cases (*p* = 0.0146, Fisher´s exact test). Thus, the introduction of a performance pay component has a statistically significant positive impact on the provision behavior of the medical students across all stylized routine cases as graphically shown in Figure 4 (*p* = 0, Fisher´s exact test). The individual behavior analysis for medical students confirms the result that they are much more profit oriented than hospital physicians (see Table A11). Moreover, the option to revise the decision after having seen an overview of all decisions is used by a similar proportion of medical students and physicians in part 1, but more frequently by the medical students than the physicians in part 2 (see Table A10). Furthermore, the revision by the students leads to the selection of the profit maximizing alternative instead of the patient optimal one in 80% of the cases.

We also conducted control treatment condition with medical students in which the monetary benefits go to real patients outside the lab. In part 1 with DRG remuneration, we find that medical students do not behave statistically significantly different to the control group of medical students with patient benefits (*p* = 0.1414, Fisher´s exact test). Comparing treatment behavior of medical students with patient benefits with hospital physicians, we find that the latter still behave in a more patient oriented way (*p* < 0.0001, Fisher´s exact test). Furthermore, we find optimizing behavior for medical students with real patient benefits in the sense that they choose statistically significantly more patient optimal alternatives in the low monetary baseline DRG incentive cases than in the high incentive ones (*p* = 0.0435, Fisher´s exact test). This changes with the introduction of the performance pay component since its impact on patient optimal behavior is consistently and statistically significantly positive (*p* < 0.0001, Fisher´s exact test). Comparing medical students with real patient benefits with hospital physicians, we find that the latter provide statistically significantly more patient optimal choices in both parts (part 1: *p* = 0.0354; part 2: *p* < 0.0001, Fisher´s exact test; also see Table A9 for an overview of the *p*-values for all Fisher´s exact tests for the entire subject pool). Finally, we check for the robustness of our main results running logit regressions (see Table A12). The results confirm the statistically significant positive effect of the performance pay component on the number of patient optimal choices, which is driven by medical students choosing more patient optimal alternatives.

**Result** **3.**
*Hospital physicians behave statistically significantly more patient oriented than both groups of medical students under the DRG system and the DRG system with a performance pay component comprising bonus–malus incentives. However, we find differences in treatment patterns. While the introduction of bonus–malus incentives leads to statistically significantly more patient optimal decisions in the high monetary baseline DRG incentive cases only for hospital physicians, both groups of medical students statistically significantly increase their number of patient optimal decisions across all stylized routine cases.*


## 5. Discussion and Conclusions

In this paper we analyze the effects of introducing a performance pay component with bonus–malus incentives to a simplified version of the current German Diagnosis Related Groups (DRG) system. For this, we impose a sequential design with a medical framing. In contrast to previous research, our stylized routine cases are presented in a medical context and always include a patient optimal and profit maximizing alternative. Our subjects pool consists of real hospital physicians and medical students.

Our results show that given a simplified version of the current German DRG remuneration system in part 1 hospital physicians choose the patient optimal alternative in 74% of all stylized routine cases. These results are in line with empirical evidence finding relatively high levels of patient orientation for physicians [52,59]. While the introduction of a performance pay component with bonus–malus incentives increases the proportion of patient optimal choices to 84% on aggregate, the increase is not statistically significant. However, at treatment case level, we find statistically significant changes towards more patient optimal behavior for cases with high monetary baseline DRG incentives to choose the profit maximizing alternative. Note that this in contrast to the findings for the South Korean PP component as they find continuous improvements in the selective therapeutic areas, e.g., acute myocardial infarction or C-sections, in which the incentives were introduced [21,22,23]. However, as noted before there are systematic differences between the South Korean and the German hospital remuneration system. In South Korea, the hospital remuneration with fee-for-service was directly replaced by a DRG system with bonus–malus incentives based on treatment quality. Hence, inferring a causal relationship between the bonus–malus incentive and the quality of care is difficult.

Even though medical students behave qualitatively similarly to hospital physicians, they choose statistically significantly less patient optimal alternatives under the simplified DRG system in part 1, i.e., 38%, and statistically significantly increase the proportion of patient optimal choices to 63% with the introduction of the bonus–malus incentives in part 2. At treatment case level, this statistically significant increase holds for all stylized routine cases. These results are robust towards introducing monetary patient benefits. The results also confirm other experiments that find a positive effect for student subject pools and performance bonus payments [43,45,46]. While Brosig-Koch et al. [58] investigate subject pool differences between physicians and medical students, we are the first to analyze a change of payment system at within subject level for medical students and real hospital physicians. In contrast to Brosig-Koch et al. [58], we find statistically significant differences between medical students and physicians which even remain with the introduction of a performance pay component with bonus–malus incentives. Hence, our results highly suggest that further experimental research investigating remuneration changes at within subject level should acknowledge that whether the change in payment scheme has a statistically significant effect on physician treatment behavior crucially depends on the initial level of patient orientation that is statistically significantly higher for real physicians. 

Overall, our results indicate that whether the introduction of a performance pay component with bonus–malus incentives to the (German) DRG system has a positive effect on the quality of care or not particularly depends on the monetary incentives implemented in the DRG system as well as the type of participants and their initial level of patient orientation. 

For policy makers, our results suggest that adding a performance pay component with a bonus–malus incentive to the German DRG system may not achieve its goal of quality improvement across all treatment cases and rigid effort should be put into designing effective performance incentives for the right treatment cases. Given our specific parametrization, we find that more money is needed to achieve one patient optimal choice for real physicians. Moreover, our experimental design including stylized routine cases may serve as a wind tunnel study for investigating payment incentives in the inpatient care sector. Future research should investigate deeper into design aspects such as varying the level incentives, the frequency of payments, or the type of performance measure, i.e., absolute and relative, as well as combining financial incentives with public quality reporting.

However, note that that the external validity of our results is limited. By using a within-subject design, we are able to identify individual behavioral changes that ceteris paribus result from introducing a performance pay component with bonus–malus incentives. Therewith, we contribute by complementing the respective field evidence that faces difficulties disentangling the effect of performance pay components from other confounders. However, such a high control of the decision environment requires one to abstract away from the field environment. In the real world decisions made, e.g., involve some form of uncertainty about the monetary outcome. Hence, future experimental research should gradually increase the realism of the decision scenario by adding, e.g., uncertainty about the performance pay component or making the latter relative and thus competitive.

## Figures and Tables

**Figure 1 ijerph-17-08320-f001:**
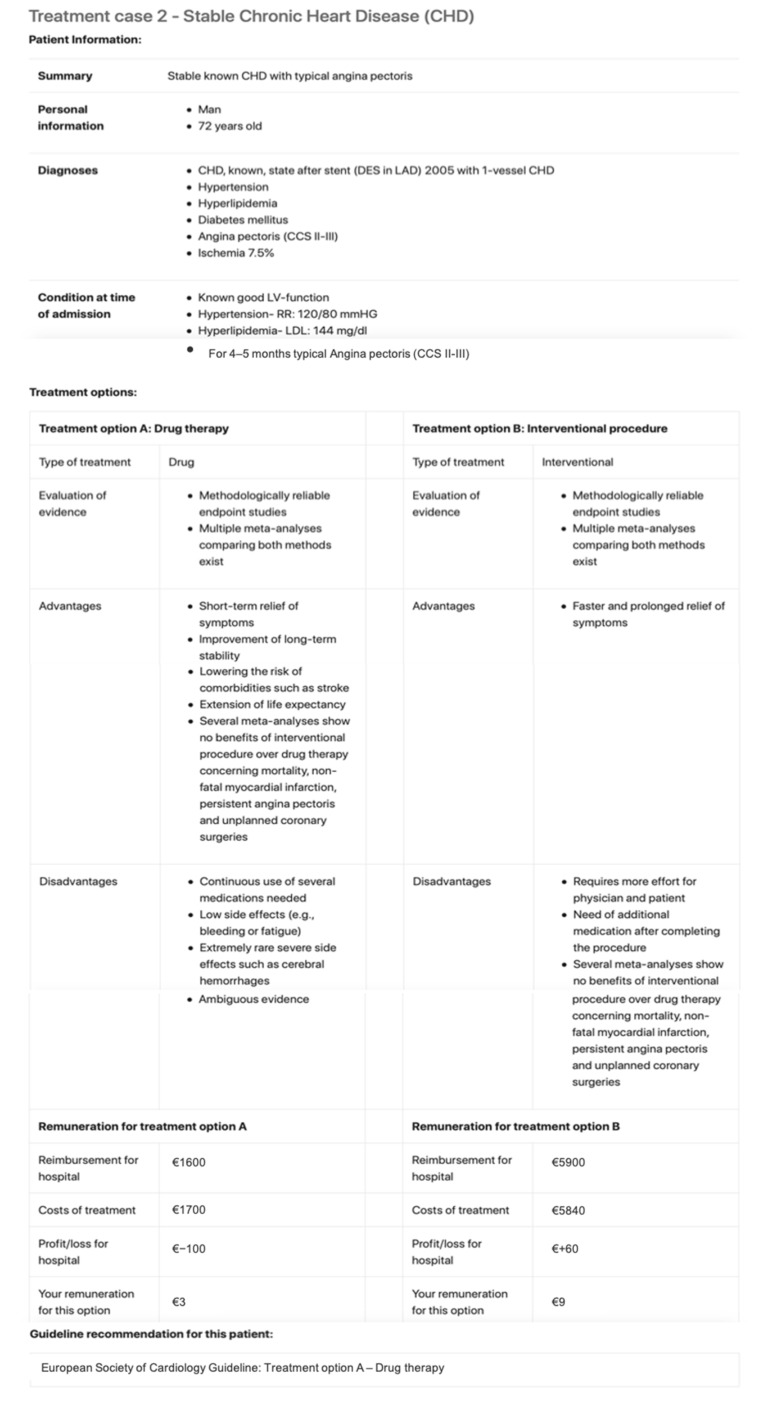
Example of stylized routine case.

**Figure 2 ijerph-17-08320-f002:**
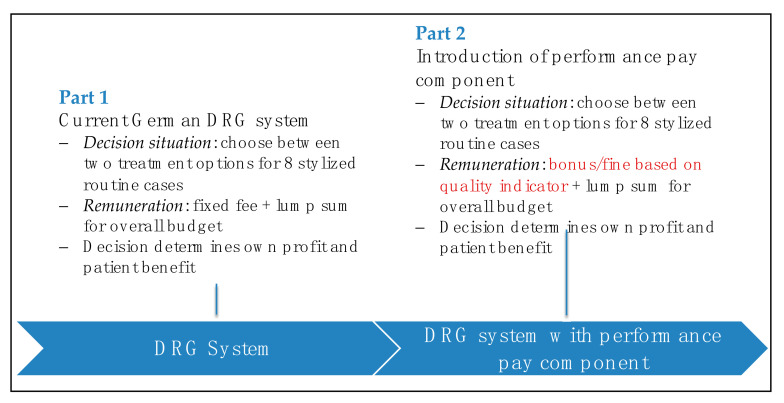
Summary of experimental design and payment incentives.

**Figure 3 ijerph-17-08320-f003:**
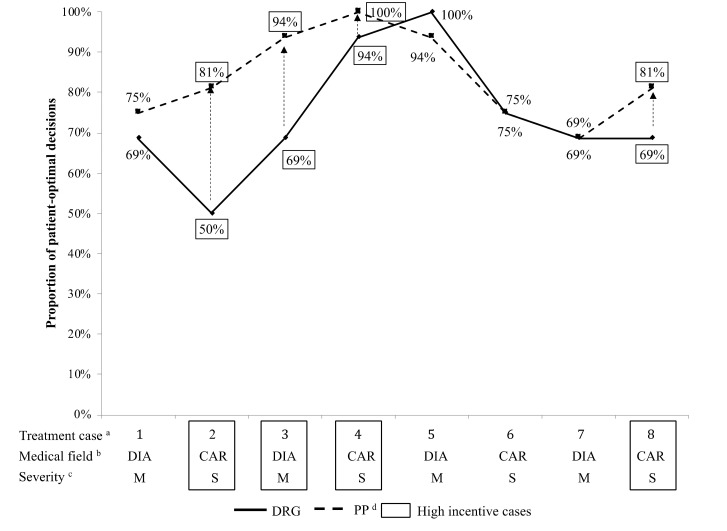
Hospital physicians’ proportions of patient optimal choices in part 1 (DRG) and part 2 (performance pay); ^a^ Stylized routine cases are displayed in order as shown in experiment; ^b^ DIA: diabetological case, CAR: cardiological case; ^c^ M: moderate case, S: severe case; ^d^ PP: performance pay.

**Figure 4 ijerph-17-08320-f004:**
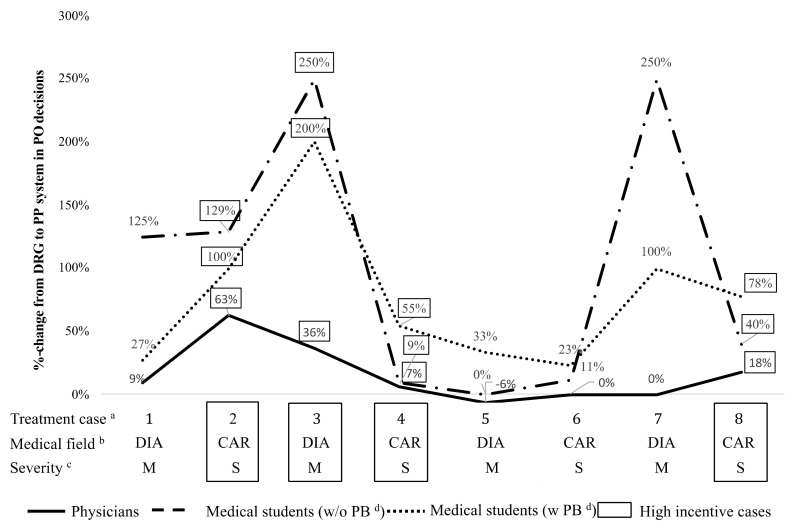
Increase of proportions of patient optimal decisions by subject pool in part 2 (performance pay); ^a^ Stylized routine cases are displayed in order as shown in experiment; ^b^ DIA: diabetological case, CAR: cardiological case; ^c^ M: moderate case, S: severe case; ^d^ PB: patient benefit, PO: patient optimal.

**Table 1 ijerph-17-08320-t001:** Treatment Conditions.

Treatment	No. of Hospital Physicians	No. of Medical Students	Total
DRG-PP/Physician	16	-	16
DRG-PP/Student	-	19	19
DRG-PP/Student+Patient	-	21	21
Total	16	40	56

Note: DRG: Diagnosis Related Groups; PP: performance pay.

**Table 2 ijerph-17-08320-t002:** Sample Characteristics for hospital physicians and medical students.

Sample Characteristics	w/o Patient Benefits	w Patient Benefits
Hospital Physicians	(*n* = 16)	n/a
Age (mean, std.dev.)	43.94 (10.17)	n/a
Gender		
% female	31.3%	n/a
Specialty		
% cardiologist	50.0%	n/a
Job level		
% physicians w/budget responsibility	68.8%	n/a
Practice years (mean, std.dev.)	15.25 (9.94)	n/a
Self-reported attitudes		
Altruism (mean, std.dev.)	16.44 (2.34)	n/a
Medical Students	(*n* = 19)	(*n* = 21)
Age (mean, std.dev.)	25.58 (5.17)	23.62 (1.80)
Gender		
% female	78.9%	76.2%
Semester (mean, std.dev.)	8.79 (2.94)	8.43 (2.77)
Self-reported attitudes		
Altruism (mean, std.dev.)	15.58 (1.98)	16.52 (2.42)

Note: w/o Patient Benefits: Treatment without donation to Christoffel Blindenmission, w Patient Benefits: Treatment with donation to Christoffel Blindenmission.

**Table 3 ijerph-17-08320-t003:** Proportions of patient optimal choices by stylized routine cases and subject pool in part 1 (DRG).

Treatment Case ^a^	1	2	3	4	5	6	7	8		
Medical Field ^b^	DIA	CAR	DIA	CAR	DIA	CAR	DIA	CAR		
Severity ^c^	M	S	M	S	M	S	M	S		
**Subject Pool**	**% of Patient Optimal Choices**								**Mean**	***p*-Value ^d^**
Physician (*n* = 16)	69%	50%	69%	94%	100%	75%	69%	69%	74%	
Student (*n* = 19)	21%	37%	21%	58%	68%	47%	21%	26%	38%	<0.0000
Student+ Patient ^e^ (*n* = 21)	52%	38%	19%	52%	71%	62%	29%	43%	46%	<0.0000

^a^ Stylized routine cases are displayed in order as shown in experiment; ^b^ DIA: diabetological case, CAR: cardiological case; ^c^ M: moderate case, S: severe case; ^d^ Note that the stated *p*-values are calculated with Fisher´s exact tests comparing the distributions of patient optimal and profit maximizing choices between the subject pools, i.e., Physician vs. Student and Physician vs. Student+Patient; ^e^ Student+Patient is the treatment with medical students in which the patient benefit is displayed in monetary terms and the amount donated to the Christoffel Blindenmission.

**Table 4 ijerph-17-08320-t004:** Proportion of patient optimal choices by treatment case and subject pool in part 2 (performance pay).

Treatment case ^a^	1	2	3	4	5	6	7	8		
Medical field ^b^	DIA	CAR	DIA	CAR	DIA	CAR	DIA	CAR		
Severity ^c^	M	S	M	S	M	S	M	S		
**Subject Pool**	**% of Patient Optimal Choices**								**Mean**	***p*-Value ^d^**
Physician (*n* = 16)	75%	81%	94%	100%	94%	75%	69%	81%	84%	
Student (*n* = 19)	47%	84%	74%	63%	68%	53%	74%	37%	63%	0.0009
Student+ Patient (*n* = 21)	67%	76%	57%	81%	95%	76%	57%	76%	73%	0.0354

^a^ Stylized routine cases are displayed in order as shown in experiment; ^b^ DIA: diabetological case, CAR: cardiological case; ^c^ M: moderate case, S: severe case; ^d^ Note that the stated *p*-values are calculated with Fisher´s exact tests comparing the distributions of patient optimal and profit maximizing choices between physicians and both student subject pools.

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
