# Peer review of "Performance Pay in Hospitals: An Experiment on Bonus–Malus Incentives"

_ijerph, 2020, doi:10.3390/ijerph17228320_

Round 1

Reviewer 1 Report

The new version are massively improved and I only have a few minor issues.

1) Give full form of DRG at first occasion outside abstract, i.e. page 2 row 58 and no do not present full form again in section 2 (page 3, line 95).

2) At page 2, row 74-82, e.g. patient optimal at row 75, you present new expressions that has not mentioned previously in the section which makes it hard to understand unless reading later parts of the text. Please revise this.

3) At page 11, row 335-339, I think that you should make it more obvious if the participant actually gets this payment or if it is only hypothetical payment. I think that you actually pay participants if they chose the profit maximazing alternative which feels very awkward for me. Please improve the text accordingly.

4) In Table 2, some very minor things, n not N as it is a sample and clarify that s.d. is standard deviation as it actually is a short form.

5) In Table 3, the table should stand by itself and for someone who has read other text poorly the alternative stud+patient needs to be clarified.

6) In Table 3, it is not understandable what the tests has done. Is it comparing row of physician with row of student or? Clarify this better.

7) At page 13, row 394 it is hard to understand what you refer to with part 1 and part 2. It would be beneficial to better clarify what these parts are so that the reader won't have to go back some pages and look closer for it.

8) Use "statistically significant" and not only "significant" to clarify that it is significant based on statistics. Should be self-obvious that you refer to statistical but still good to be as clear as possible.

9) There is still very little discussion about how useful the results are in terms of generalizability. There is a weakness in that it is only a hypothetical set-up and even if results are linked to previous research and seems to be in line the paper would benefit from a better explanation on why this results can be trusted. I think that it would be from the benefit for the authors to better prove that this weakness can be handled adequately.

Author Response

See pdf attached.

Reviewer 2 Report

Thanks for the revision

Author Response

Dear Revewier, We are happy we could convince you with our revised version and thank you very much for your efforts.

Reviewer 3 Report

I appreciate changes and can accept th etext in its present form.

Author Response

Dear Reviewer, We are happy we could convince you with our revised version and thank you very much for your efforts.

This manuscript is a resubmission of an earlier submission. The following is a list of the peer review reports and author responses from that submission.

Round 1

Reviewer 1 Report

The authors are taking on an important and tricky to evaluate topic.

The approach might be a feasible one but I have big difficulties in judging things due to a lack of clarity and some important points.

To start with I find it hard to understand the design. Medical students and physicians are approached to participate. They should choose between two alternatives for a patient. If I understood things correct the choice is purely based on information and not by a real meeting, at least does this seem very obvious from the manuscript. The participant choose between performance pay and patient optimal alternative.

There is no information about the ethics in the study. If I have understood things right are the participants given economical incentives for participation. This part about the compensation really puzzles me. Are the payment only hypothetical or are the participants getting this based on their decision? If they get the money and it is not a real patient why would they not then of their own greed choose the payment alternative? If they do not get the payment themselves why would they not tend to respond the patient optimal alternative as it seems to be the most honorable choice? If they get paid for participating then how would this affect the selection of study subjects, i.e. physicians and students? I see no reasoning around this.

The hypothetical set-up for the study certainly gives raise to a lot of questions. Still these issues are not debated at all in the discussion. The weaknesses of the hypothetical questions and how that can affect the results must be handled deeply. I lack some clear statement on how you can verify that the participant choose as they would do in clinical practice.

Also the non-random selection of participants and how they might not be representive is very important to discuss. As it was difficult to recruit cardiologists and diabetologists it might be related to the conducted research and thus affect the validity of the results a lot.

Another thing I don't understand is if you consider one of the options, patient optimal or profit maximization to be the correct one. It really needs more of a clarity in whether they choose the right option or how the objective of the study should be seen. Does the government with the profit maximizing component want the best quality and at the same time reward the physicians for taking such decision even if means that sometimes less quality will be chosen? The rationale behind the two examples to choose between is unclear to me.

The text where the experiments are explained are long and can be shortened and especially be more easily understood. Maybe a box or a table can be used to really easily clarify what this to treatment cases are and how they should be understood. I have done my best to read the text slowly and carefully but I still have problems understanding how they are supposed to work.

These are my main overall criticism that I think need to be resolved before considering to accept the manuscript for publication.

Besides this there are some more straightforward concerns I have:

1) Clearly explain how people were invited. Were it by annonces and everyone who responded participated? Or did you approach people and a number of people participated? The recruitment process must be very well explained and it is important that this information is given in the abstract as well. Now it is not even mentioned how many that responded.

2) The abstract must be improved with actual results. Now the reader cannot make any judgement themselves but are completely relying on other to make valid conclusions.

3) There are too many abbreviations in the text. It makes it harder to read than necessary. I see for instance no use of using PP, PO and PM as it requires myself and other readers to go back and check what did this mean when it actually is just two words that they are written instead of.

4) In background it needs to be explained what malus incentives are. This might be well-known term in economics but outside that field it is not. The paper should be possible to read smoothly also by public health experts.

5) Counts should be added in tables. They should be self-explanatory and now there is not even information about how many that participated in the study and this is of course even more important when the sample size is so low.

6) Are all participants responding to both experiments? Please add a sentence or two that clarifies this.

7) A general advice is to go through the text and think about the message that are meant to be given in each paragraph and rewrite it accordingly. Currently there are much text but still some crucial information is lacking as I mentioned in this report.

Before the above issues are clarified it is difficult to judge the value of the conducted study. The potential validity issues that have not been discussed might mean that it is not really possible to trust the results. That being said, I am only unable to make a judgement on this which does not mean that the study and the analyses were not performed in a way that makes them valuable.

Author Response

Please see the attachment."

Reviewer 2 Report

The authors conduct a controlled laboratory experiment to examine how performance pay influences treatment decisions in a prospective payment scheme. Since performance pay has rarely been implemented in practice, this experimental investigation is an important step to guide future health care policy. Nevertheless, I have some remarks on how to further strengthen the paper before publication.

  • There should be an example in the introduction on how bonus or malus incentives could be applied in the actual hospital setting. In the experiment, it directly affects the physician. But is this realistic? How does it work in South Korea? We need to know from the introduction that we should care about this!

  • The authors should describe in more detail how DRG systems work and why there is a trade-off at all. The reimbursement for the hospital is based on the average costs that hospitals have. Differences in profit hospitals can make on each case hence depend on hospital characteristics (e.g. specialization). It might well be that certain hospitals have a larger monetary gain from surgeries compared to pharmaceutical-based care but they are a special case. Hafsteinsdottir & Siciliani (2010) provide a theoretical argument for differences in care based on financial incentives when DRGs are split. In case the authors assumed that both treatment options are reimbursed by one DRG then they should definitely show examples where in the German DRG system they found such a setting.

  • Why should physicians care about how financially beneficial their treatment choices are? This needs to be explained in the context of the experiment and related to the real world (it is not usual that there is any kind of financial incentive for physicians working in hospitals to save costs / create revenue). Reference (46) in the paper only deals with heads of medical units who are only consulted in complex cases (which is not the case here as these seem to be quite standard, even medical students can deal with them).

  • Quality and outcomes are deterministic in the experiment (physicians just choose it). How realistic is this? Please discuss this more carefully. In particular when outcomes are stochastic, how would the possible gains from bonus payments outweigh the risks of malus payment?

Minor comments:

  • Footnote 5 should mention Australia because basically, that’s where G-DRGs come from.
  • Line 91: Here is the first occurrence where (CAR) should be used.
  • The authors should name the DRGs used to calculate the values in the experiment.
  • Line 244: Does a significance test make sense when there is no variation?
  • Line 281: One “this” is enough.
  • Table “Experimental Parameters and Annotations”: Is “DRG fee” really the correct word? From the perspective of the physician, it is the reimbursement.
  • Appendix A.5: “Ihr Budget” should be translated.
  • The authors claim, that the funders had no role in the design of the study. However given that a pharmaceutical company funded the research, is it a coincidence that the drug therapy is the better option?

References:

Hafsteinsdottir, Elin Johanna Gudrun, und Luigi Siciliani. 2010. „DRG Prospective Payment Systems: Refine or Not Refine?“ Health Economics 19(10):1226–39.

Reviewer 3 Report

Interesting study, but some extra work is needed. I have following core suggestions:

a/ Introduction should not serve as literature review part (minimum references should be included in it).

b/ Most critical comment - literature review part on performance payment and its problems (in general and in health care) is missing. Such text is necessary, findings are fully in line with existing knowledge on the topic, but this is not visible from the text now (conclusions also suffer from this weakness).

c/ Another critical comment - the authors should better explain "bonus-malus incentives" and to provide minimum needed characteristics of participating doctors (are all of them from public health sphere, or private medicine is included?)

d/ Are there any other factors (like legal risks connected with choosing the treatment path) influencing decisions of doctors? The discussion and conclusions should be based on interdiciplinarity principle.
